# Determining Digital Representation and Representative Elementary Volume Size of Broken Rock Mass Using the Discrete Fracture Network–Discrete Element Method Coupling Technique

**Xiao Huang** [1,2,3,*], **Siyuan Li** [1], **Jionghao Jin** [1] **and Chong Shi** [4]

1   Department of Civil Engineering, Zhejiang Ocean University, Zhoushan 316022, China;
    siyuan.li@zjou.edu.cn (S.L.); jinjionghao@zjou.edu.cn (J.J.)
2   Ocean College, Zhejiang University, Zhoushan 316021, China
3   Zhejiang Jinggong Steel Building Group Co., Ltd., Shaoxing 312030, China
4   Institute of Geotechnical Engineering, Hohai University, Nanjing 210024, China; scvictory@hhu.edu.cn
*   Correspondence: 2022144@zjou.edu.cn

**Abstract:** Obtaining the digital characterization and representative elementary volume (REV) of broken rock masses is an important foundation for simulating their mechanical properties and behavior. In this study, utilizing the broken surrounding rock of the main powerhouse at the Liyang pumped storage power station as an engineering background, a three-dimensional fracture network generation program is first developed based on the theories of discrete fracture network (DFN) and discrete element method (DEM). The program is then integrated with a distinct element modelling platform to generate equivalent rock mass models for broken rock masses based on the DFN–DEM coupling technique. Numerical compression tests are conducted on cylindrical rock specimens produced using the proposed modelling approach, aiming to determining the REV size of the target rock masses at the Liyang power station. A comparative validation is also performed to examine the REV result obtained from the proposed approach, which adopted a REV measuring scale index (RMSI) to determine the REV size. Results indicate that the organic integration of DFN simulation techniques and DEM platforms can effectively construct numerical models for actual broken rock masses, with structural surface distributions statistically similar to the real ones. The results also show that the REV size of the investigated rock masses determined by the cylindrical rock models is 5 m × 10 m, which aligns with the size determined by the cubic rock models, as the target cubes show the same height as the cylindrical specimens. This study provides a model and parameter basis for the numerical calculation of the mechanical behavior of broken rock mass.

**Keywords:** broken rock mass; statistics; DFN–DEM coupling; REV; Monte Carlo

## 1. Introduction

As a geological formation that has undergone prolonged natural processes, rock masses preserve distinct traces of geological tectonic activities. Within various geological regions, the rock masses manifest diverse quantities and types of structural surfaces. For broken rock masses that are characterized by a high-dispersion structure due to extensive jointing, fissuring, and faulting, the discontinuity and anisotropy of the structure become more pronounced [1,2]. In engineering practices involving broken rock masses, such as tunnels at near-fault regions and side slopes in highly weathered rock formations, the randomness of discontinuities, the variability in rock mass parameters, and the uncertainty associated with surrounding rock loads pose potential risks to the safety of the construction projects [3,4]. Therefore, to enhance the construction safety of such engineering practices, it is essential to conduct in-depth investigations on the mechanical properties and behavior of broken rock masses.

Due to the haphazard structural surfaces within the broken rock masses, their mechanical properties are particularly complex and challenging to be predicted quantitatively. Even with the implementation of large-scale rock mechanics tests on-site, it is only possible to obtain local mechanical parameters of limited-sized rock blocks or joint surfaces. Directly measuring the macroscopic or statistically meaningful mechanical properties and parameters of larger-scale rock masses is unattainable through experiments [5,6]. Therefore, building upon experiments and theoretical analyses, numerous researchers have employed numerical methods to construct computational models of fractured rock masses and carry out relevant numerical mechanical tests. For example, the equivalent rock mass technology, which combines a three-dimensional fracture network simulation technique and discrete element platform, has been extensively employed in the construction of fractured rock mass models at both indoor and engineering scales [7–12]. Meanwhile, the parameters of the equivalent rock masses, such as stiffness, deformation modulus, Poisson's ratio, internal friction angle, trace length, and fracture aperture, have also been adopted to estimate the REV size of fractured rock masses [13–22]. However, for broken rock masses with a more intricate distribution of fractures, limited research has been conducted on the determinations of their digital characterization and REV. Moreover, such studies have not fully captured the structural characteristics of the real rock masses. The random fracture network based on the probabilistic model description is a local macro-fracture, which only reflects some of the main structural features of the broken rock mass, and the structural surface simulation technology, which is closer to the real rock mass structure, still needs to be developed.

With the continuous development of pumped storage power station projects in the North and Southeast coastal regions of China, broken surrounding rock masses of large underground cavern groups, which serve as the main structural component of pumped storage power stations, have become increasingly prevalent in practical engineering projects. In this paper, based on the relevant theories of discrete fracture network (DFN) and discrete element method (DEM), the distribution characteristics of structural surfaces in the broken surrounding rock masses of a specific hydropower station in Southeast China, i.e., the Liyang pumped storage power station, is statistically analyzed. By employing the Monte Carlo method, a three-dimensional fracture network automatic generation program is developed based on the DFN–DEM coupling approach, which can construct the equivalent rock mass models for broken rock masses at a certain site. Using the proposed program, cylindrical rock mass models with different scales are generated to analyze the size of the RVE of the broken rock mass at the Liyang pumped storage power station. The numerical results are validated through a comparison with the results obtained from an equivalent rock mass cubic model, employing the RMSI with identical parameters.

## 2. Characterization of Random Fracture Networks in Broken Rock Masses

The simulation technique for rock mass fracture networks was originally introduced by the British scholar Samaniego [10]. This method allows the reproduction of the structural surface of a realistic rock mass based on the statistical distribution of geometric parameters measured from real rock surfaces, and the Monte Carlo method is employed to implement this reproduction process on a computer platform [23]. In the past two decades, many scholars have conducted in-depth research on this technique and developed corresponding programs to reproduce the rock structural surface, such as the Fracman program, Blocks program, and Geofrac program [24]. These programs are generally based on the Monte Carlo method to generate rock mass fracture networks, and the basic process of Monte Carlo random simulation technology can be summarized as (1) conducting on-site joint geological surveys to obtain basic geological parameters of rock fracture samples, such as fracture density, attitude, spacing, and aperture ratio; (2) performing statistically analyses on the geological parameters of the fracture samples and fitting corresponding probability density and distribution functions; and (3) using the Monte Carlo method to generate random fracture networks that statistically match the actual distribution of the rock mass

fractures based on specific distributions of random numbers. The implementation process of random fracture network simulation is shown in Figure 1.

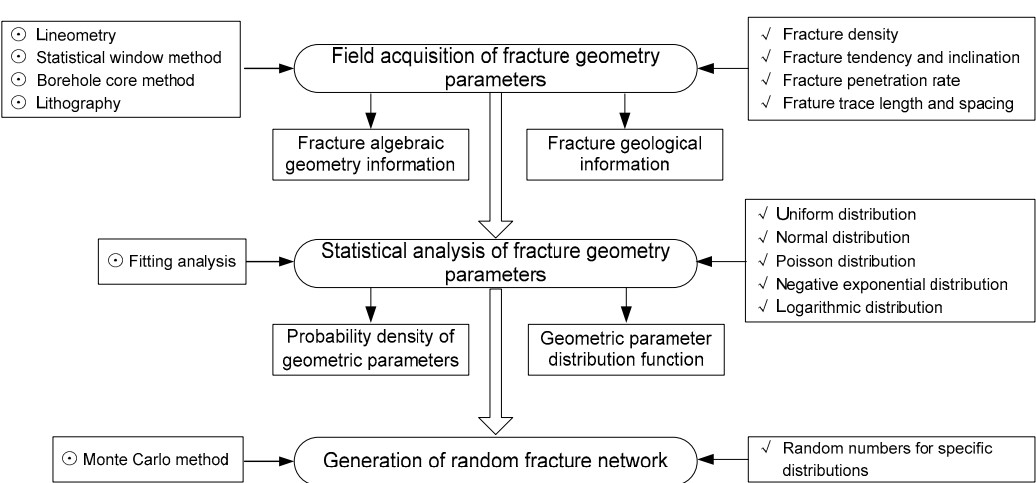

**Figure 1.** Generation process of random fracture network.

Figure 1 summarizes the generation process of the random fracture network adopted in this research to simulate broken rock masses, which contains three main steps, namely, (1) the field acquisition of fracture geometry parameters, (2) statistical analysis of the parameters, and (3) generation of random fracture network for the rock masses. In this paper, the broken surrounding rock of the main powerhouse at the Liyang pumped storage power station is adopted as the research object to provide fracture geometry parameters. In step 2, a self-developed program is employed to conduct statistical analysis on the fracture geometric parameters within a predefined range of the target rock (i.e., a statistical window). In step 3, a three-dimensional fracture network automatic generation program is developed to generate the random fracture networks for the target rock masses. The obtained network diagram can be converted into a DXF file that is readable by 3DEC.

*2.1. On-Site Research of Structural Surface*

Geological parameters of the dominant structural surfaces in a research area are the key sample data for simulating random fracture networks. In the literature, the methods for experimentally obtaining the geological parameters mainly include manual precise measurement methods (such as rock outcrop line measurement and statistical window methods), borehole core method, a photography or videography method, and three-dimensional laser scanning technology.

In this study, the statistical window method [25] is adopted for on-site investigation of structural surfaces. A "Geological Structural Surface Statistical Window" program is developed to statistically analyze the geometric parameters of structural surfaces obtained for the on-site investigation, which will be introduced in the following sections. Compared to other methods, the primary characteristic of the statistical window method is to designate a rectangular range on the rock outcrop surface as a "statistical window"; the statistical relationship between the geometric parameters of the structural surfaces within this window and the parameters of the window itself can thus be established. A schematic diagram illustrating the collection of structural surface data on a target rock mass using the statistical window method is shown in Figure 2.

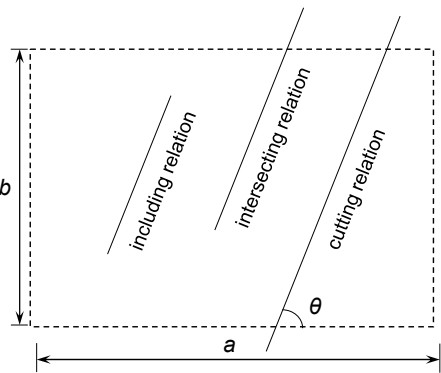

**Figure 2.** Schematic diagram of rock mass structural plane collected by statistical window method.

*2.2. Mathematical Description of Stochastic Model for Geometric Parameters of Structural Surfaces*

Previous studies showed that the geometric feature parameters of the structural surface of rock mass obey a certain form of probability distribution. By fitting and analyzing the collected data of the structural surfaces of the broken rock mass, a statistical probability model for each geometric feature parameter of these surfaces can be established.

2.2.1. Shape and Size of Structural Surfaces

In the literature, most studies assumed a circular or elliptical shape for the structural surfaces for simplicity. Nonetheless, some research considered a polygonal shape [25]. Zhao et al. [26] proposed the use of a circular shape to describe structural surfaces in blocky rock masses and a rectangular shape for those in layered rock masses. In homogeneous rock masses, it is reasonable to represent a single structural surface by a circle or an ellipse. However, in broken rock masses, due to the criss-crossing and interacting cracks, the shape of the structural surfaces is polygonal. Therefore, this paper uses planar quadrilateral (rectangle) to simulate the shape of the structural surfaces in broken rock masses.

The dimensions of structural surfaces are generally described by the fracture trace length, which is the intersection of the fracture surface with the exposed rock surface. As of now, it is not possible to directly measure the size of fracture surface in three-dimensional space under field conditions. The size can only be derived from the fracture trace length obtained from a field survey based on relevant theories. From the perspective of probability statistics, the distribution models of fracture trace length mainly include lognormal distribution [27] and negative exponential distribution [28]. The density functions of these two distributions are shown in Figure 3. It can be seen from Figure 3a that, when adopting log-normal distribution to describe the fracture trace length, i.e., when *x* equals the fracture trace length, the probability of occurrence for very short fracture trace lengths approaches zero. This can be attributed to the limitations of existing observation methods in measuring fine cracks, which may not accurately represent real cases, as a large number of laboratory experiments have observed numerous micro-cracks in the rock masses. To better describe the distribution of fracture trace length in broken rock masses, this study adopted the negative exponential distribution, as illustrated in Figure 3b. The figure demonstrates a notable increase in the number of cracks as the fracture trace length decreases, aligning more closely with the experimental results.

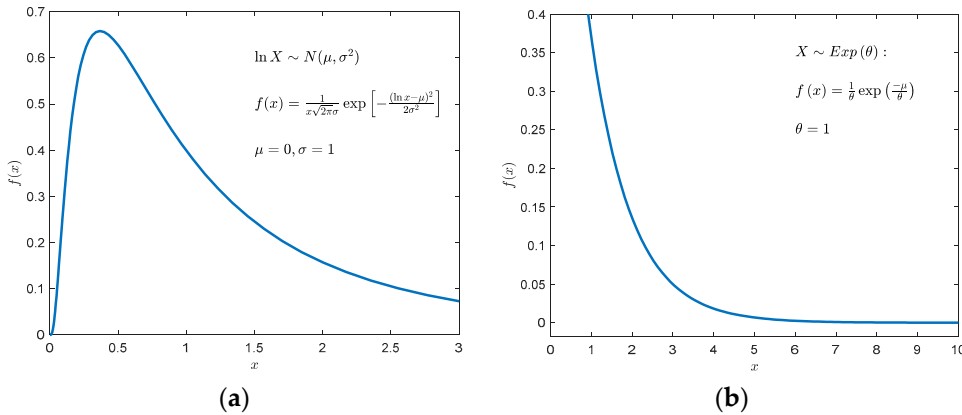

**Figure 3.** Density functions for log-normal and negative exponential distributions: (**a**) log-normal distribution; (**b**) negative exponential distribution.

To establish the dimension of structural surfaces, a rectangular statistical window $a \times b$ is first set up on the rock outcrop face (see Figure 2). Assuming that the numbers of structural surfaces whose traces are in cutting, intersecting, and containing relationships with the statistical window are $N_0$, $N_1$ and $N_2$, respectively, and the angle between the traces of this group of structural surfaces and the length direction of the statistical window is $\theta$, the average trace length of this group of structural surfaces, $l$, can be expressed as:

$$l = [ab(1 + N_0/N) - N_2/N]/[(1 - N_0/N + N_2/N)(aA + bB)] \tag{1}$$

where $N = N_0 + N_1 + N_2$, $A = \int_{\theta_l}^{\theta_u} f(\theta) \sin\theta d\theta$, and $B = \int_{\theta_l}^{\theta_u} f(\theta) \cos\theta d\theta$. $f(\theta)$ is the probability density function of $\theta$, and $\theta_l$ and $\theta_u$ are the lower and upper values of $\theta$, respectively.

Villaescusa and Brown provided a relationship between the structural surface trace length and the diameter of circular structural surface [29]. Based on this relationship, the mean and variance values of the structural surface diameter, denoted as $D$ and $\sigma_D{}^2$, can be expressed as the following equation.

$$\overline{D} = \frac{3}{2}\left(\frac{3}{8}\right)^6 (\pi\bar{l})^5 / \left[(\bar{l})^2 + \sigma_l{}^2\right]^2, \ \sigma_D{}^2 = 3\left(\frac{3}{8}\right)^{10}(\pi\bar{l})^8 / \left[(\bar{l})^2 + \sigma_l{}^2\right]^3 \tag{2}$$

There is also a conversion relationship between the diameter of the circular structural surface and the length of the rectangular structural surface $B$, i.e., $\overline{B} = \sqrt{\pi} \cdot \overline{D}/2$. However, this relationship is subject to the prerequisite that the length and width of the rectangular structural surface are independent of each other and both conform to the negative exponential distribution of parameter $B$. The substitution of the negative exponential distribution parameter $1/\lambda = B$ into $x_i = -[\ln(1 - R_i)]/\lambda$ yields the random distribution of rectangular edge length, in which $R_i$ is an obeyed (0, 1) interval on the uniform distribution. The side lengths of the rectangular structural surface, $a_i$ and $b_i$, can thus be calculated as:

$$a_i = -\sqrt{\pi/4D}\ln(1 - R_i), \ b_j = -\sqrt{\pi/4D}\ln(1 - R_j) \tag{3}$$

### 2.2.2. Occurrence of Structural Planes

There are two main methods for describing the occurrence of structural planes [30]. One approach involves expressing the occurrence of structural planes in the normal direction. The other uses the strike and dip to describe the occurrence. The latter uses dip azimuth $\varphi_d$ and dip angle $\theta_d$ to represent the occurrence of structural plane, which is widely used in geological engineering, as shown in Figure 4.

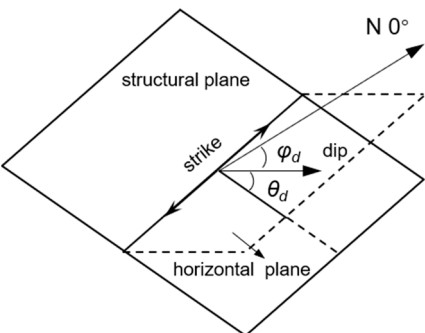

**Figure 4.** Tendency and dip angle representation of structural plane.

The commonly used probability distributions of structural plane occurrence (parameters including dip azimuth angle $\varphi_d$ and dip angle $\theta_d$) include Bingham distribution, Fisher distribution, and bivariate normal distribution. Among these distributions, the first two regard the azimuth and inclination of the structural plane as two independent random variables, while the last one considers the correlation between the two parameters. Consider a set of two-dimensional random variables $\varphi_d$ and $\theta_d$ with sample means $\overline{\varphi_d}$ and $\overline{\theta_d}$, variances $\sigma_\varphi^2$ and $\sigma_\theta^2$, respectively, and a correlation coefficient $\rho$ between the two random variables. The probability density function is as follows:

$$f\left(\frac{\varphi_d - \overline{\varphi_d}}{\sigma_\varphi}, \frac{\theta_d - \overline{\theta_d}}{\sigma_\theta}, \rho\right) = \frac{1}{2\pi\sigma_\varphi\sigma_\theta\sqrt{1-\rho^2}} \exp\left[\frac{\frac{2\rho(\varphi_d-\overline{\varphi_d})(\theta_d-\overline{\theta_d})}{\sigma_\varphi\sigma_\theta} - \left(\frac{\varphi_d-\overline{\varphi_d}}{\sigma_\varphi}\right)^2 - \left(\frac{\theta_d-\overline{\theta_d}}{\sigma_\theta}\right)^2}{2(1-\rho^2)}\right] \tag{4}$$

The joint probability distribution of the two variables can be expressed as follows:

$$P(\varphi_d \leq u, \theta_d \leq v) = \int_{-\infty}^{\frac{v-\theta_d}{\sigma_\theta}} \int_{-\infty}^{\frac{u-\varphi_d}{\sigma_\varphi}} f\left(\frac{\varphi_d - \overline{\varphi_d}}{\sigma_\varphi}, \frac{\theta_d - \overline{\theta_d}}{\sigma_\theta}, \rho\right) d\left(\frac{\varphi_d - \overline{\varphi_d}}{\sigma_\varphi}\right) d\left(\frac{\theta_d - \overline{\theta_d}}{\sigma_\theta}\right) \tag{5}$$

When the dip azimuth angle $\varphi_d$ and dip angle $\theta_d$ are independent of each other, i.e., $\rho = 0$, the bivariate normal distribution in Equation (4) degenerates into a one-dimensional normal distribution. Substituting the mean and mean square deviation of the occurrence parameters $\varphi_d$ and $\theta_d$ into the one-dimensional normal distribution $x_i = \sigma N_i + \mu$, the azimuth angle $\varphi_{di}$ and dip angle $\theta_{di}$ of the structural plane that satisfy the normal distribution can be obtained:

$$\varphi_{di} = \sigma_\varphi N_i + \varphi_d, \quad \theta_{di} = \sigma_\theta N_i + \theta_d \tag{6}$$

### 2.2.3. Density of Structural Planes

The density of structural surfaces can generally be classified as line density, areal density, and volume density based on spatial dimensions, with interconvertibility among them. Line density of structural planes is the reciprocal of the spacing between structural planes, while areal density and volume density refer to the numbers of centerpoints of structural plane traces per unit area and unit volume of the rock mass, respectively. Among them, areal density is more widely used to evaluate the density of structural surfaces in geotechnical engineering, and the commonly used methods for estimating areal density include the Kulatilake estimation method [31] and the Mauldon estimation method [32].

Based on the correlation between sample window traces and the window in the statistical window method, as shown in Figure 2, Kulatilake and Wu [31] proposed a formula for calculating the areal density $E(\rho_a)$ of structural planes:

$$E(\rho_a) = \frac{\sum_{i=1}^{N_2} [P_0(W)]_i + \sum_{i=1}^{N_1} [P_1(W)]_i + \sum_{i=1}^{N_0} [P_2(W)]_i}{ab} \tag{7}$$

where $a$ and $b$ denote the length and width of the statistical window, respectively; $N_0$, $N_1$, and $N_2$ represent the number of structural plane traces in the cutting, intersecting, and enclosing relationships with the statistical window, respectively; and $P_0(W)$, $P_1(W)$, and $P_2(W)$ are the probabilities of the midpoint of surface trace corresponding to the above three relationships being inside the statistical window.

Similarly, the expression of areal density of structural planes proposed by Mauldon [32], $\lambda$, can be described by the following equation based on the statistical window method:

$$\lambda = (N - N_2 + N_0)/2ab = (2N_0 + N_1)/2ab \tag{8}$$

By comparing Equations (7) and (8), it is evident that Mauldon's method for estimating the areal density of rock masses within a statistical window is more straightforward than Kulatilake's method, as the Kulatilake method necessitates calculations involving the probability density functions of the trace length in the study area. Based on the statistical analysis results from the Mauldon method, the study adopts the two-parameter negative exponential distribution model as the probabilistic statistical model for the surface density of structural planes in broken rock masses. The two-parameter negative exponential distribution can be expressed as:

$$y = me^{-nx} \quad (x \geq 0, \quad m > 0, \quad n > 0) \tag{9}$$

where $m$ and $n$ are the model parameters.

As the density of structural planes increases, the spatial distribution of fractures become more complex, which leads to more permutations and combinations of random geometry parameters that can be generated by the Monte Carlo method.

### 2.2.4. Center Point and Vertex of a Rectangular Structural Surface

The position of a rectangular structural plane can be determined by the coordinates of its center and four vertices. In the literature, it is generally assumed that the center point of a structural plane to follow a three-dimensional Poisson distribution [28]. Based on the principle of Poisson's process, the determination of the spatial coordinates of the center point of a structural surface can be outlined as follows: (1) divide the three-dimensional space region $V$ into $N$ independent subregions $V_i$ with equal volumes ($i$ = 1, 2, 3..., $N$); (2) assume the center points of the $N_i$-th structural surface are randomly and uniformly distributed in the $V_i$ subregion; (3) assume the range of the $V_i$ sub-region to be $x_{i1} \leq x \leq x_{i2}$, $y_{i1} \leq y \leq y_{i2}$, and $z_{i1} \leq z \leq z_{i2}$, and assume a variable $R_i$ following a uniform distribution on the interval (0, 1). The coordinates of the structural plane's center point, ($x_{ci}$, $y_{ci}$, $z_{ci}$), can thus be expressed as:

$$x_{ci} = x_{i1} + (x_{i2} - x_{i1})R_i, y_{ci} = y_{i1} + (y_{i2} - y_{i1})R_i, z_{ci} = z_{i1} + (z_{i2} - z_{i1})R_i \tag{10}$$

If the upper and lower sides of the rectangular structural plane have independent and identically distributed lengths, the lengths $a_i$ and $b_i$ of the rectangular structural plane can be obtained from Equation (3), the dip azimuth angle $\varphi_{di}$ and dip angle $\theta_{di}$ of the structural plane can be obtained from Equation (6), and the coordinates of the center point of the structural plane ($x_{ci}$, $y_{ci}$, $z_{ci}$) can be obtained from Equation (10). Based on these conditions, the vertex coordinates of the rectangular surface can be solved as follows.

In Figure 5, assuming that the unit vector of side **AB** of the rectangular structural surface is $N_1$, the unit vector of side **BC** is $N_2$, and the unit normal vector to the structural surface is $N_i$ (perpendicular to plane ABCD), where $N_i = (\sin\theta_{di}\sin\varphi_{di}, \sin\theta_{di}\sin\varphi_{di}, \cos\theta_{di})$ can be calculated based on the relationship between the normal vector of the structural plane and the occurrence (dip azimuth and dip angle). Since **AB** is perpendicular to both

$N_i$ and $z$-axis, and **BC** is perpendicular to both $N_i$ and $N_1$, the following relationships can be established:

$$N_1 = [(0, 0, 1) \times N_i]/\mathrm{mod}[(0, 0, 1) \times N_i] = (-\cos\varphi_i, \sin\varphi_i, 0) \tag{11}$$

$$N_2 = (N_i \times N_1)/\mathrm{mod}(N_i \times N_1) = (-\sin\varphi_i\cos\theta_i, -\cos\varphi_i\cos\theta_i, \sin\theta_i) \tag{12}$$

where $(0, 0, 1)$ represents the unit vector in the $z$-axis direction. In addition to these unit vectors. the coordinates of vertex A, i.e., $(x_a, y_a, z_a)$, can also be obtained from the vector relations in Figure 5, which can be expressed as:

$$OA = (x_a - x_{ci}, y_a - y_{ci}, z_a - z_{ci}) = OE + EA \tag{13}$$

$$OE = -b_i N_2/2, \qquad EA = -a_i N_1/2 \tag{14}$$

The substitution of Equations (11) and (12) into Equations (13) and (14) gives the coordinate of vertex $A(x_a, y_a, z_a)$ can be obtained. Similarly, the coordinates of the rest vertices, namely $B(x_b, y_b, z_b)$, $C(x_c, y_c, z_c)$, and $D(x_d, y_d, z_d)$, can be calculated and expressed as the following equation:

$$x_j = x_{ci} + (b_i\sin\varphi_i\cos\theta_i + \theta_i\cos\varphi_i)/2,\ y_j = y_{ci} + (b_i\cos\varphi_i\cos\theta_i - \theta_i\sin\varphi_i)/2,\ z_j = z_{ci} - (b_i\sin\theta_i)/2 \tag{15}$$

where $j$ represents the four vertex indicators, $a$, $b$, $c$, and $d$.

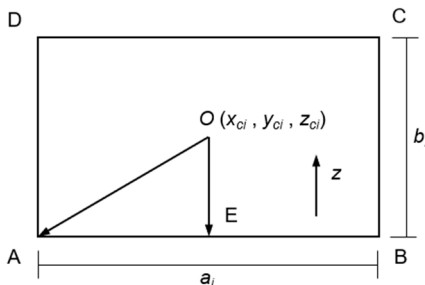

**Figure 5.** Relationship between center point and vertex of a rectangular structural surface.

2.2.5. Degree of Aperture of a Structural Plane

The degree of aperture is the vertical distance between the rock walls on both sides of an open structural surface. Due to the undulating nature of the fracture walls, the degree of aperture of a structural plane is different at different points. By establishing a coordinate system "$xoy$" on the mid-plane of a structural surface, as shown in Figure 6, the aperture degree of a certain point with coordinates $(x, y)$ can be expressed as a function, $E(x, y)$. Taking the average value of these $E(x, y)$ for the points throughout the fracture surface gives the average degree of aperture for the structural plane $d$. If the side lengths of a rectangular fracture surface are $L_x$ and $L_y$, respectively, $d$ can be expressed as:

$$\bar{d} = \frac{1}{L_x L_y} \int_0^{L_x} \int_0^{L_y} E(x, y)dxdy \tag{16}$$

The aperture degrees of a structural plane mainly obeys a negative exponential distribution [31] or a log-normal distribution [33]. If the average value of the aperture degrees in a fracture group is $d$ and obeys a negative exponential distribution, and the variable $R_i$ satisfies an uniform distribution within the interval $(0, 1)$, the aperture degree of any structural plane $d_i$ can be expressed as:

$$d_i = -d\ln(1 - R_i) \tag{17}$$

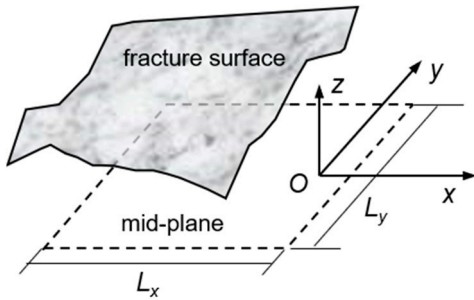

**Figure 6.** Solution model of structural plane aperture.

### 2.3. Generation of Random Fracture Network Model

After obtaining the distribution model of the geometric parameters of a structural plane through on-site rock mass investigation and statistical analysis, the random fracture network for the structural plane can be generated using computer simulation techniques. Such generation process is opposite to the on-site measurement and statistical process. Specifically, based on the survey and statistical results, a probability function of the geometric parameters of the investigated structural plane is first established. A series of random numbers that conform to specific distribution functions is then generated using approaches including the Monte Carlo method to replace the geometric parameters of the structural plane. A structural plane network diagram is finally generated to represent the rock mass in the study area.

The Monte Carlo method is a widely employed numerical technique based on random numbers in the simulation of fracture networks [26]. Its fundamental concept involves approximating solutions to real-world problems through simulations and statistical experiments on random variables. The key to this method lies in generating random numbers based on specific probability distributions. Various method, such as displacement instruction addition, square-centering approach, and congruence method, can be used for generating random numbers, typically following a uniform distribution [34]. To obtain random numbers that adhere to the desired distribution, random sampling of uniform distributed random numbers is required according to the established distribution functions. Some commonly used methods to generate random numbers for different probability density functions are summarized in Table 1.

**Table 1.** Random number generation of different probability density functions.

| Distribution Type | Probability Density Function | Random Numbers | Fissure Applications |
|---|---|---|---|
| Uniform distribution | $f(x) = \begin{cases} 1/(b-a) & a \leq x \leq b \\ 0 & x < a,\, x > b \end{cases}$ | $x = a + (b-a)R$ | Midpoint location |
| Negative exponents distribution | $f(x) = \begin{cases} \lambda \exp(-\lambda x) & a \leq x > 0 \\ 0 & x \leq 0\, \lambda > 0 \end{cases}$ | $x = -\ln(1-R)/\lambda$ | Trace length |
| Fisher distribution | $f(x) = \dfrac{\eta \sin\theta \exp(\eta\cos\theta)}{\exp\eta - \exp(-\eta)}$ | $x = \arccos\left[\dfrac{\ln(1-R)}{\eta} + 1\right]$ | Dip angle |
| Log-normally distributed | $f(x) = \dfrac{1}{\sqrt{2\pi}\sigma x} \exp\left[-\dfrac{(\ln x - \mu)^2}{2\sigma^2}\right]$ | $u = \begin{cases} \cos(2\pi R_1)\sqrt{-2\ln R_2} & R_3 \geq 0.5 \\ \sin(2\pi R_1)\sqrt{-2\ln R_2} & R_3 < 0.5 \end{cases}$ $x = \exp(\mu + \sigma\mu)$ | Trace length/Aperture |
| Poisson distribution | $P(x = k) = \dfrac{\lambda^k e^k}{k!}\ k = 0,1,\ldots \lambda > 0$ | $F(n) = \sum\limits_{k=0}^{n} \dfrac{\lambda^k e^{-\lambda}}{k!}$ $if\ F(n) \leq R \leq F(n+1)\ then\ x = n$ | Density |
| Normal distribution | $f(x) = \dfrac{1}{\sqrt{2\pi}\sigma} \exp\left[-\dfrac{(x-\mu)^2}{2\sigma^2}\right]$ | $u = \begin{cases} \cos(2\pi R_1)\sqrt{-2\ln R_2} & R_3 \geq 0.5 \\ \sin(2\pi R_1)\sqrt{-2\ln R_2} & R_3 < 0.5 \end{cases}$ $x = \mu + \sigma\mu$ | Aperture |

## 3. Modeling of Broken Rock Masses Based on the Coupled DFN–DEM Approach

Based on the pertinent theories of DFN simulation and DEM, the fractured surrounding rock of the main powerhouse at the Liyang pumped storage power station is adopted as an illustrative example. The actual geometric parameters of the structural surface of the rock masses, as well as their respective probability distributions, are obtained first, and the Monte Carlo method is then employed to simulate the corresponding random fracture network. Finally, through the implementation of the coupled DFN–DEM approach, an equivalent rock mass model is constructed.

### 3.1. Statistics of Structural Plane Geometric Parameters

To acquire the authentic geometric parameters of the fractured surrounding rock in the main powerhouse of the Liyang pumped storage power station, we conducted an exhaustive statistical analysis using on-site geological survey data. The data sources encompassed the "Geological Sketch Display of the Roof Arch of the Main Powerhouse", "Geological Sketch Display of the Upstream Wall of the Main Powerhouse", "Geological Sketch Display of the Downstream Wall of the Main Powerhouse", and "Geological Sketch Display of the North End Wall and South End Wall of the Main Powerhouse". Employing the statistical window method allowed for a comprehensive statistical analysis. Figure 7a–c show the placement of three geological statistical windows situated on the roof arch, upstream sidewall, and downstream sidewall of the main powerhouse (excluding large faults during this process).

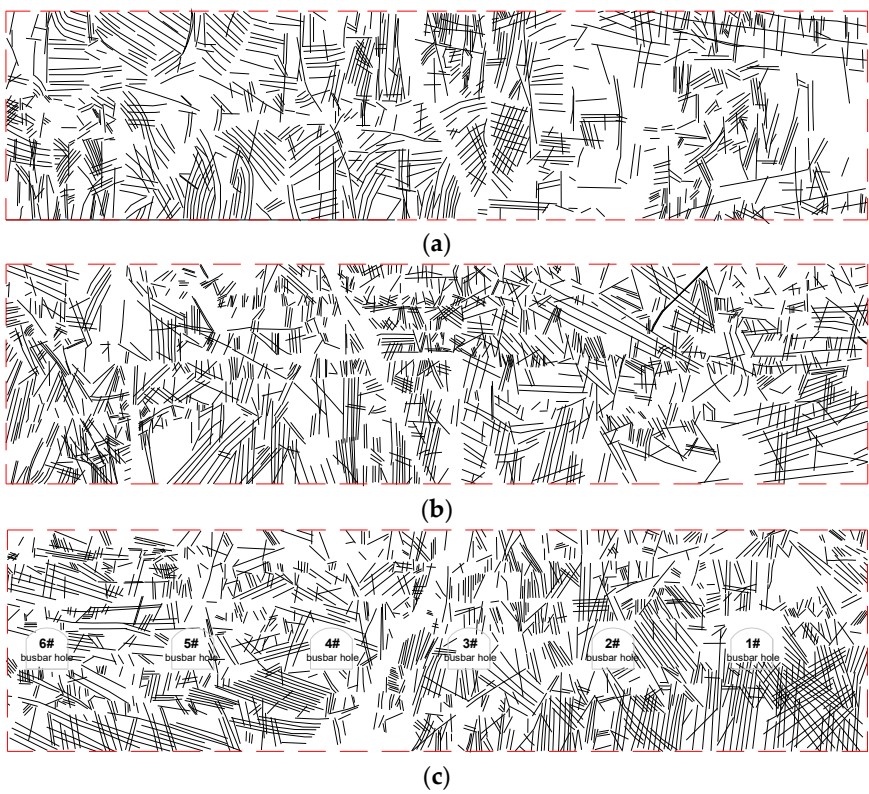

**Figure 7.** Structural plane statistical windows in main powerhouse area: (**a**) roof arch statistical window; (**b**) upstream sidewall statistics window; (**c**) downstream sidewall statistics window.

A custom program called "Geological Statistical Window Analysis" is developed based on the quantitative relationship between the geological geometric parameters of structural surfaces within the statistical windows. By importing a DXF file into the "Geological Statistical Window Analysis" program, one can obtain geometric parameters such as the count, dip angle, trace length, line density, areal density, and volume density of structural surfaces within the statistical windows.

Upon organizing the result files, the actual geometric parameters of structural surfaces within the rock masses of the arch roof, upstream sidewall, and downstream sidewall of the main powerhouse are collected, as presented in Table 2. The following can be observed from the table: (1) In comparison to the upstream and downstream sidewalls, the fractures in the arch roof rock mass exhibit the maximum length, averaging 4.99 m. However, they have the fewest fractures per unit area of rock mass (i.e., 0.15 fractures per $m^2$). (2) When contrasted with the arch roof and downstream sidewall, the fractures in the upstream wall rock mass display a more evenly developed length (variance of trace length equals 2.85 $m^2$), and the highest number of fractures per unit length of rock mass (0.87 fractures per meter). (3) The average trace length and trace length variance of structural surfaces in the surrounding rock of the main plant building are 4.30 m and 3.45 $m^2$, respectively. The average dip and dip variance of structural surfaces are $51.42°$ and $(27.93°)^2$, respectively. The number and length of fractures per unit area of rock mass are 0.23 and 0.97 $m^{-1}$, respectively. The number of fractures per unit length of rock mass is 0.58 $m^{-1}$, and the fracture area per unit rock mass is 1.87 $m^{-1}$.

**Table 2.** Analysis results of geological statistical window of surrounding rock of main powerhouse.

| Geologic Statistics Window Locations | Roof Arch | Upstream Wall | Downstream Wall | Average |
|---|---|---|---|---|
| Number of structural faces/strips | 1049 | 1238 | 1421 | 1236 |
| Average trace length of fractures/m | 4.99 | 3.76 | 4.15 | 4.30 |
| Trace length variance/$m^2$ | 4.00 | 2.85 | 3.50 | 3.45 |
| Fracture average dip angle/° | 43.26 | 56.32 | 54.69 | 51.42 |
| Dip angle variance/$(°)^2$ | 29.76 | 27.01 | 27.02 | 27.93 |
| Quantity areal density (number of fractures per unit area)/$m^{-2}$ | 0.15 | 0.29 | 0.26 | 0.23 |
| Fracture density (length of fracture per unit area)/$m^{-1}$ | 0.75 | 1.07 | 1.10 | 0.97 |
| Fracture line density (number of fractures per unit length)/$m^{-1}$ | 0.31 | 0.87 | 0.55 | 0.58 |
| Three-dimensional fracture density (fracture area per unit volume)/$m^{-1}$ | 2.40 | 1.22 | 1.99 | 1.87 |

### 3.2. Construction of Discrete Fracture Networks

Based on relevant theories in structural statistics of rock masses, this research develops a tool named "Random Fracture Network Simulation Program for Fractured Rock Masses". This program encompasses a fracture data input module, a fracture network generation module, and a fracture network output module, offering the following features and advantages: (1) Real-time selection of parameter distribution functions and generation of fracture networks based on the input geometric parameters of structural surfaces. (2) The output result file can be inputted into some other software for further analysis, providing ease of connection with program interfaces. (3) A user-friendly interface that requires minimal compilation of data files, effectively reducing the likelihood of errors. The existing approaches for simulating broken rock masses generally require numerical simulations to obtain the probabilistic distributions of the structural surfaces' geometric parameters [30]. In the proposed method, the optimal probabilistic distribution functions for the geometric parameters can be automatically acquired without additional simulations, which can significantly accelerate the simulation process.

The simulation process of the program is illustrated in Figure 8. Initially, within the fracture data input module, a database of geometric parameters for structural surfaces is established. The number of structural plane groups, as well as the probability distribution forms for the geometric parameters of each structural plane group, can thus be determined. Next, employing the Monte Carlo principle, the program calculates the center point, vertex coordinates, and geometric parameters for each fracture to complete the simulation for each group of structural surfaces. To optimize the structural plane network, fractures that are in close proximity or have very small angles are merged or deleted; fractures beyond the model dimensions are truncated and removed. Finally, after completing the simulation of the structural surface network, the program outputs result in required formats.

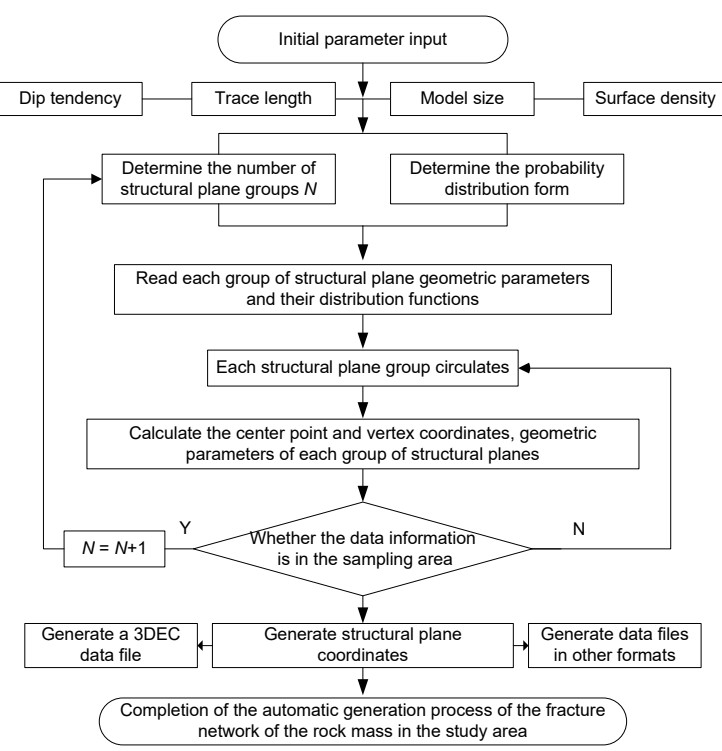

**Figure 8.** Flow chart of random fracture network simulation of fractured rock mass.

A random fracture network with specified dimensions of a rock mass can be generated by inputting the statistical data of geological features and model dimensions of the structural surfaces into the "Fracture Rock Random Fracture Network Simulation Program". However, the generated result files are not directly applicable. Therefore, a file conversion interface program is developed to convert the generated random fracture network into a DXF file that is readable by 3DEC. In this example, the structural surface geometric parameters of the rock mass around the main powerhouse, as listed in Table 3, are inputted into the "Fracture Rock Random Fracture Network Simulation Program". The result files are imported into the conversion program, which gives a 3DEC-readable random fracture network, as illustrated in Figure 9.

**Table 3.** Physical and mechanical parameters of rock block.

| Density /g·cm$^{-3}$ | Bulk Modulus /GPa | Shear Modulus /GPa | Cohesion /MPa | Internal Friction Angle/° | Tensile Strength /MPa |
|---|---|---|---|---|---|
| 2600 | 24.6 | 12.5 | 11.3 | 50 | 3.58 |

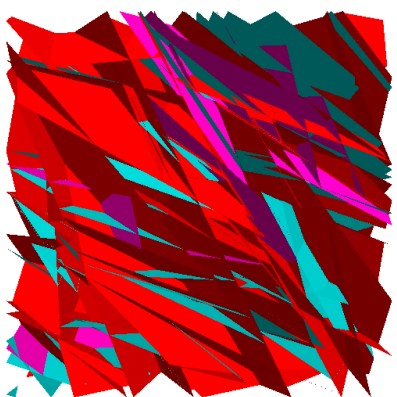

**Figure 9.** Three-dimensional random fracture network (a cube with side length of 20 m).

### 3.3. Construction of Equivalent Rock Mass Models Using DFN–DEM Coupling Technique

The discrete element method [35], also known as the distinct element method, is a numerical method based on the mechanics of discontinuous media. Its fundamental principle, grounded in Newton's second law, involves the consideration of contact surfaces or interfaces between small blocks (particles) that constitute as system, allowing for a more accurate simulation of the mechanical behavior of the blocks and contact interfaces. Currently, widely employed programs for discrete element simulations include the two-dimensional discrete element program UDEC (Universal Distinct Element Code), block discrete element program 3DEC (3-dimensional Distinct Element Code), and particle flow codes (PFCs). These program tools are particularly well-suited for analyzing the response of fractured rock systems or assemblies of discontinuous blocks under static or dynamic loading conditions.

The DFN–DEM coupling technique is a method that integrates DFN techniques for modelling fractured rock masses with the DEM for numerical experiments to determine the mechanical properties of fractured rock masses. In this study, the programs introduced in Section 3.2 are utilized as the DFN generator, while the numerical discrete element analysis is performed using the 3DEC 7.0 software, chosen for its efficiency in modeling and analyzing large-scale systems. The integration of the DFN simulation tools with 3DEC 7.0 software facilitates the utilization of the DFN-generated random fracture network to cut the discrete element blocks, constructing a numerical model with statistically similar fracture distributions to the actual rock masses, as demonstrated in Figure 10.

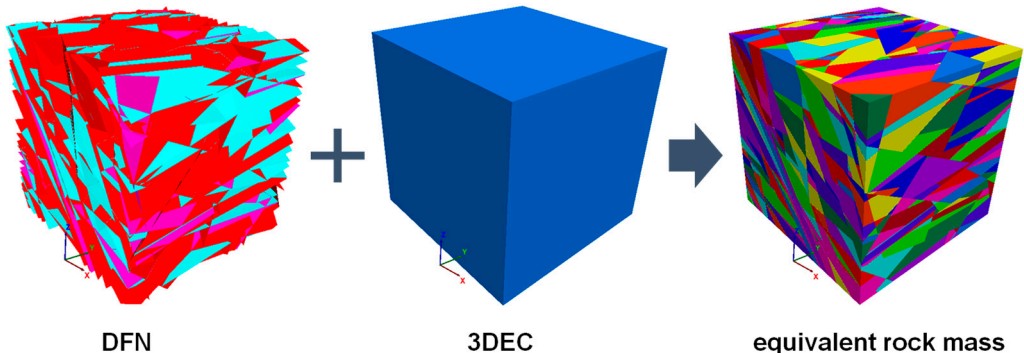

**DFN** **3DEC** **equivalent rock mass**

**Figure 10.** Rock mass model based on DFN–DEM coupling.

### 4. REV Size of Broken Rock Mass

Given the abundance of random fractures in broken rock masses, determining the REV sizes and investigating the size effect on the mechanical parameters of the rock masses become essential aspects in the analysis of broken rock masses. The fundamental approach for obtaining the REV of a rock mass is to conduct numerical experiments on cubic rock models at different scales, while in laboratory experiments, cylindrical rock specimens with preset aspect ratios (height-to-diameter ratios) are often adopted [36,37]. In the literature, there is relatively limited research on conducting REV size analysis using cylindrical numerical models. Therefore, in this section, based on the equivalent rock mass technology, we adopted cylindrical models of different scales to investigate the REV of brokens rock masses.

### 4.1. Establishment of Numerical Models

In order to study the REV of broken rock masses, a series of uniaxial compression numerical tests are carried out in this numerical example. While maintaining the same aspect ratio, namely, a height-to-diameter ratio equal to 2, different cylindrical specimens with varying diameters are used to analyze the trends of mechanical parameters (e.g., peak compressive strength and elastic modulus) with changing diameter size. The required geometric parameters of the structural surfaces can be found in Table 3. Based on the

proposed DFN–DEM coupling method, numerical models for the cylindrical rock specimens with diameters of 1 m, 2 m, 3 m, 4 m, 5 m, 6 m, 7 m, 8 m, 9 m, 10 m, 15 m, and 20 m are established. Figure 11 shows the equivalent rock mass numerical model for these cylindrical models after one Monte Carlo experiment. Since each generation of the fracture network requires a Monte Carlo experiment, the DFNs generated from the same statistical model parameters of a structural surface are not exactly the same, even with the same model size. Therefore, for each of these 12 models, the average values of the mechanical parameters computed from five different simulations are adopted as indices for the REV scale of the model.

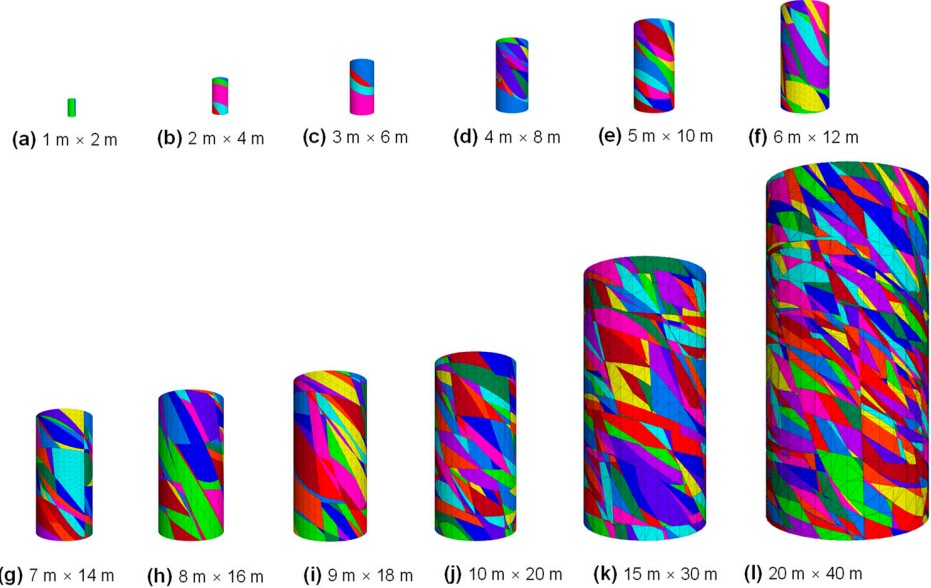

**Figure 11.** Equivalent rock mass numerical model generated by a Monte Carlo test with different scales.

In the numerical models, the rock blocks are simulated using the Coulomb–Mohr model, while the joints are represented by the Coulomb slip joint model. The physical and mechanical parameters of both the rock blocks and joints are determined through laboratory experiments [38], as summarized in Tables 3 and 4. Within the DFN models, fractures represent real joints, whereas other joints are assigned with virtual properties, simulating the mechanical behavior of intact rock blocks. To conduct uniaxial compression numerical experiments using the numerical models, rigid bearing plates are applied to the top and bottom of a model; the physical and mechanical properties of these plates are summarized in Table 5. The bottom plate is set as a fixed constraint, and the top plate applies axial compression to the model at a constant velocity (i.e., 0.001 m/s) until failure occurs. The built-in "fish" language in 3DEC is utilized to monitor and record the variation curves of axial stress and strain for the cylindrical rock model. Specifically, all nodes on the top and bottom surfaces of the model are marked, and their axial forces and axial displacements are recorded at each time step. The axial stress of the model is calculated by summing the axial forces on the top and bottom surfaces, dividing by the surface area, and calculating the average, while the axial strain of the model is computed by subtracting the bottom displacement from the top displacement and dividing the result by the original height of the cylinder.

**Table 4.** Physical and mechanical parameters of structural planes.

| Normal Stiffness/GPa | Tangential Stiffness/GPa | Tensile Strength/MPa | Cohesion/MPa | Tensile Strength /MPa |
|---|---|---|---|---|
| 30 | 15 | 0.3 | 1.1 | 30 |

**Table 5.** Mechanical parameters of rigid bearing plate.

| Density/g·cm$^{-3}$ | Bulk Modulus/GPa | Shear Modulus/GPa |
|---|---|---|
| 7800 | 300 | 250 |

### 4.2. Determination of REV

The numerical uniaxial compression tests provide stress–strain curves of the cylindrical rock specimens with different scales. To illustrate the stress–strain curves, Figure 12 shows the results from the uniaxial compression tests on the group of 3 m × 6 m cylindrical specimens, from which it can be seen that the peak axial stresses for the five specimens are 3.16 MPa, 4.46 MPa, 6.78 MPa, 3.43 MPa, and 3.84 MPa, respectively, and the elastic moduli are 7.91 GPa, 9.17 GPa, 9.67 GPa, 6.94 GPa, and 7.21 GPa, respectively. Although the parameters are not precisely identical, their close approximation to each other validates, to a certain extent, the efficacy of the proposed method for modeling fractured rock masses. The average peak compressive strength and the average elastic modulus of this group of models are 4.33 MPa and 8.18 GPa, respectively.

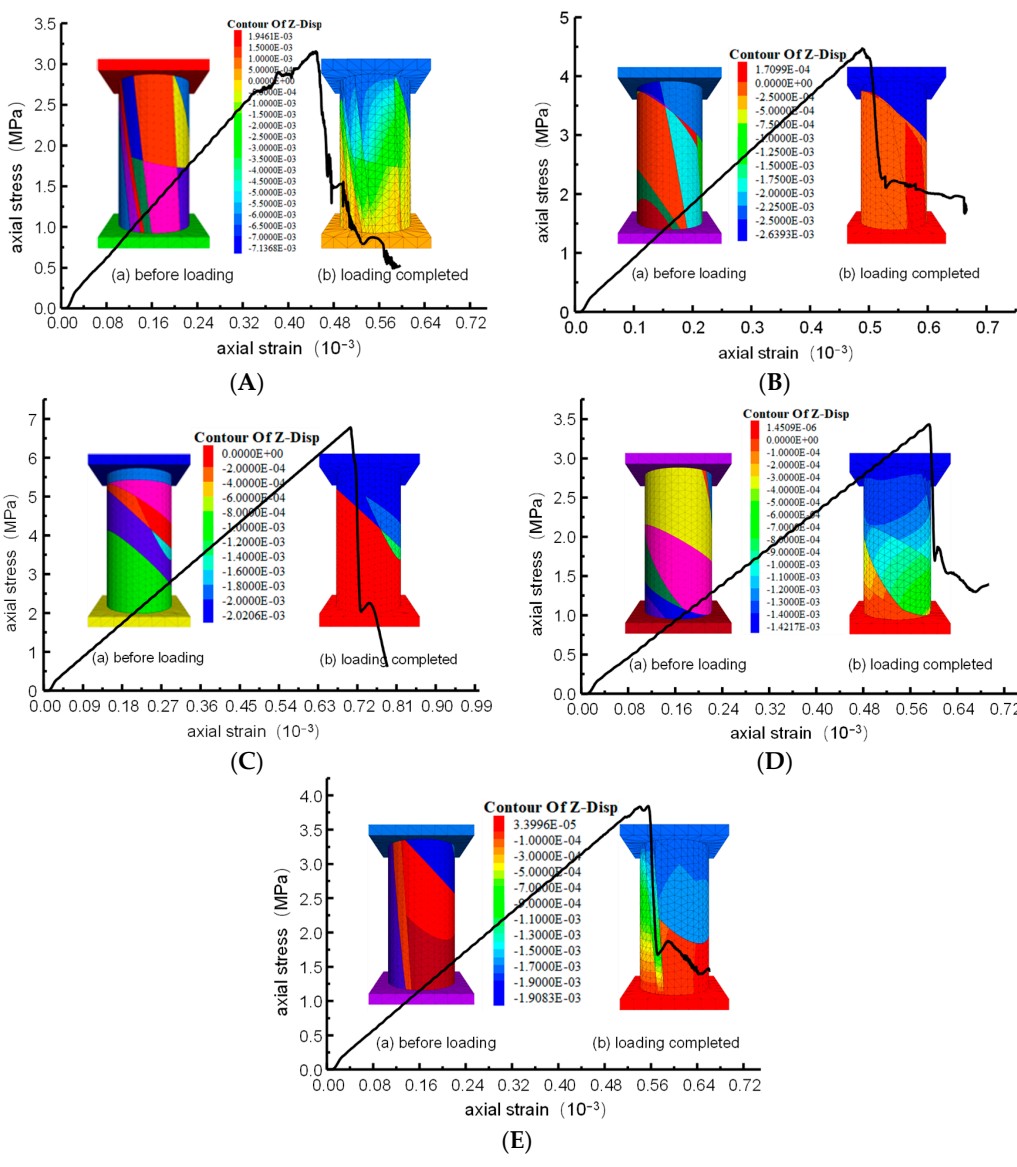

**Figure 12.** Uniaxial compression test results of cylindrical rock samples (3 m × 6 m): (**A**) the first Monte Carlo trial; (**B**) the second Monte Carlo trial; (**C**) the third Monte Carlo trial; (**D**) the fourth Monte Carlo trial; (**E**) the fifth Monte Carlo trial.

From this example, it is evident that the equivalent rock masses and their mechanical parameters generated for the same model size and structural plane geometric parameters are not identical, which can be attributed to the fact that the distributions of the simulated structural planes are only statistically similar.

The variations in peak compressive strength and elastic modulus of the rock model groups with specimen diameter of the groups are shown in Figure 13. The figure demonstrates evident size effects on the uniaxial compressive strength and elastic modulus of the broken rock masses. When the size of the equivalent rock mass is small (cylinder diameter less than 5 m), the average uniaxial compressive strength and average elastic modulus decrease rapidly with the increasing diameter of the specimens; when the diameter of the equivalent rock mass is more than 5 m, the evaluated mechanical parameters exhibit insignificant changes with increasing diameter. Therefore, the cylinder model size of 5 m × 10 m is approximately the REV size of the investigated broken rock masses.

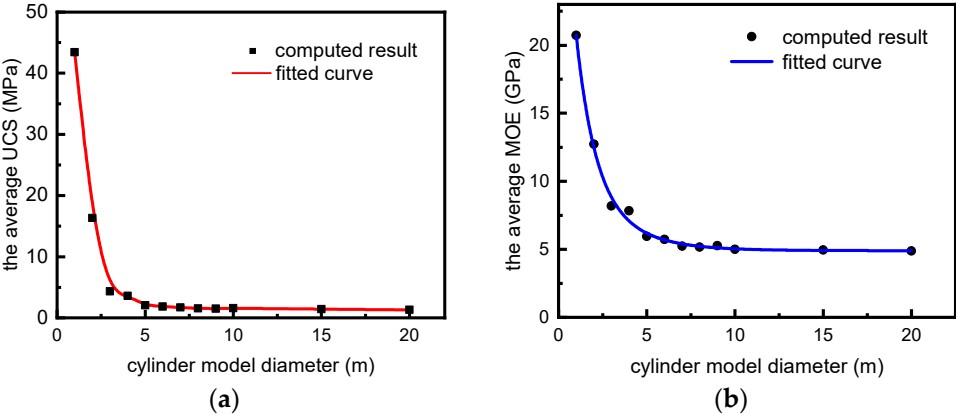

**Figure 13.** Relationship between mechanical indexes of equivalent rock mass with different sizes and diameter of cylinder model: (**a**) average uniaxial compressive strength vs. model size; (**b**) mean modulus of elasticity vs. model size.

To verify the correctness of the numerical results, a REV measurement scale index (RMSI) proposed by Zhang [39] is adopted to establish the relationships between these indices and the size length of numerical cubic rock specimens for the same rock masses at the Liyang pumped storage power station. The RMSI is defined as the projected area on a section per unit volume of the rock mass, which can be expressed as the following equation:

$$RMSI = \frac{\sum\limits_{i=1}^{m} S_i \left( \overrightarrow{N_i} \bullet \overrightarrow{N_p} \right)}{V} \tag{18}$$

where $S_i$ is the area of the $i$-th structural plane, $N_i$ is the unit normal vector of the structural plane, $N_p$ is the defined unit normal vector, and $V$ is the volume of rock mass.

Figure 14 shows the numerical cube-model for the equivalent rock masses and the relationships between the examined RMSIs and the side length of cubic models. It can be seen from the figure that, initially, when the side length is small, the REV measurement scale indices of the rock masses increase rapidly with increasing side length of the models. When the side length exceeds 10 m, the RMSIs stabilize to specific values, which indicates a side length of 10 m of the numerical cubic rock model to be the correct size for measuring the REV size of the target rock masses. This size agrees with that determined from the cylindrical rock model, which verifies the correctness of the numerical results from the proposed method.

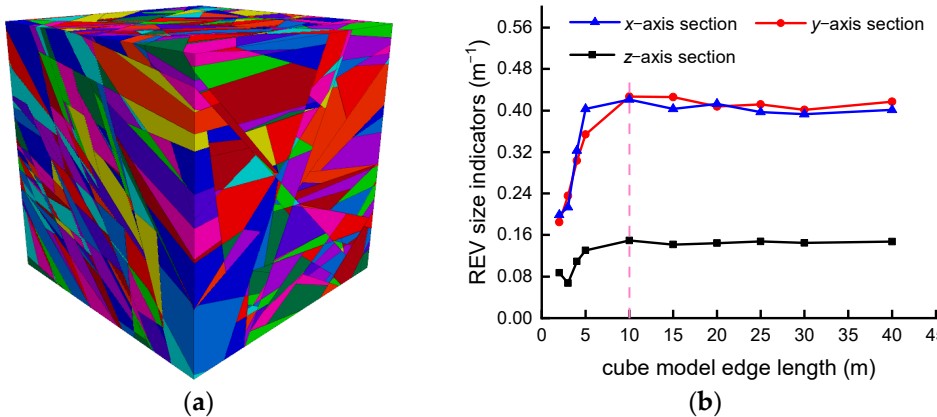

**Figure 14.** Size effect results of equivalent rock cube model: (**a**) cube-model of the equivalent rock mass; (**b**) the relationship between the REV size indices and the side length of the cube model.

## 5. Conclusions

In this paper, a numerical approach is developed for the effective generation of equivalent broken rock mass models, allowing for the investigation of mechanical behavior and the determination of REV size of given broken rock masses. A statistical program is built first to evaluate the geometry parameters of the target rock masses; the parameters are then inputted into a self-development numerical program to generate the discrete fracture network based on the Monte Carlo method. A distinct element modelling platform, namely 3DEC, is applied in conjunction with the developed programs to form a complete DFN–DEM coupling network. The performance of the proposed approach is demonstrated through a numerical example, which evaluates the mechanical parameters and REV size of the surrounding rock of the main powerhouse at the Liyang pumped storage power station using cylindrical rock models. The determined REV size is verified by the RMSI that evaluates the REV size using cubic rock model. The numerical study yields the following conclusions:

(1) The average and variance values of the structural plane in the surrounding rock of the main powerhouse are 4.30 m and 3.45 m$^2$, respectively, and the average and variance of the structural plane dip angles are 51.42° and (27.93°)$^2$, respectively. The count and length of fractures per unit area of the rock masses are 0.23 and 0.97 m$^{-1}$, respectively; the number of fractures per unit length of the rock is 0.58 m$^{-1}$; and the fracture area per unit length is 1.87 m$^{-1}$.

(2) The mechanical parameters evaluated from five cylindrical models generated using the proposed approach are quite close, which suggests good stability and effectiveness of the method.

(3) The REV size of the rock masses determined using the proposed approach is 5 m × 10 m for the cylindrical rock models. This result is consistent with the REV size estimated by the RMSI method using cubic rock models, which demonstrates the validity of using cylindrical models to evaluate the REV size of broken rock masses.

In general, the numerical approach developed in this study exhibits/demonstrates desirable effectiveness in modelling broken rock masses. Once the geometrical parameters of the fracture in the engineered rock mass are obtained, a numerical model reflecting the structural characteristics of the fractured rock mass can be constructed by using the Random Fracture Network Simulation Program to provide a basis for simulating and investigating the engineering problems of the fractured rock mass. In addition to 3DEC, some other discrete element analysis platform, such as UDEC, can work in conjunction with the developed programs for modelling large-scale mass structures and analyzing corresponding engineering problems. However, due to the difficulty in directly measuring the mechanical properties of larger-scale rock masses, the accuracy of the approach can

only be numerically evaluated. Future studies are expected to further validate the precision of numerical modelling methods for broken rock masses.

**Author Contributions:** Conceptualization, X.H. and C.S.; Funding acquisition, X.H. and C.S.; Investigation, X.H.; Software, X.H. and J.J.; Validation, X.H. and S.L.; Visualization, X.H.; Writing—original draft, X.H. and S.L.; Writing—review and editing, X.H. and J.J. All authors have read and agreed to the published version of the manuscript.

**Funding:** The work presented in this paper was financially supported by the National Natural Science Foundation of China (Grants Nos. 41831278) and the Fundamental Research Funds for Zhejiang Provincial Universities and Research Institutes (Grants Nos. JX6311041223).

**Institutional Review Board Statement:** Not applicable.

**Informed Consent Statement:** Not applicable.

**Data Availability Statement:** The data presented in this study are available in article.

**Conflicts of Interest:** Author Xiao Huang was employed by the company Zhejiang Jinggong Steel Building Group Co., Ltd. The remaining authors declare that the research was conducted in the absence of any commercial or financial relationships that could be construed as a potential conflict of interest.

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
