# Peer review of "Determining Digital Representation and Representative Elementary Volume Size of Broken Rock Mass Using the Discrete Fracture Network–Discrete Element Method Coupling Technique"

_applsci, doi:10.3390/app14020606_

Round 1
Reviewer 1 Report
Comments and Suggestions for Authors
I strongly recommend authors to read carefully all the remarks/corrections/suggestions included in the attached file, and subsequently correct properly the article for a final, acceptable form.

Comments on the Quality of English LanguageEnglish is fine and minor corrections were suggested.
Reviewer 2 Report
Comments and Suggestions for Authors
Abstract and Introduction (Sections Abstract and 1)
The abstract provides a good summary of the study but could benefit from a brief mention of the key findings and their potential implications for the field​​.
In the introduction, while the significance of studying broken rock masses is well-articulated, it would be beneficial to briefly outline the challenges and limitations of existing methods before introducing the new approach​​.
Characterization of Random Fracture Networks (Section 2)
The section on the characterization of random fracture networks is well-detailed. However, it would be useful to include a brief discussion on the limitations or challenges of using the Monte Carlo method and how these were addressed in this study​​.
Consider adding a comparison of the new DFN-DEM method with other existing methods for characterizing fracture networks to highlight its advantages or improvements.
Mathematical Description and Occurrence of Structural Planes (Sections 2.2 and 2.2.2)
This section provides a thorough explanation of the mathematical modeling of structural surfaces. It would be beneficial to the reader if you could include a practical example or a case study to illustrate how these mathematical models are applied in real-world scenarios​​.
Include a brief section on how the choice of probability distributions (e.g., Bingham, Fisher) affects the results and the rationale behind selecting particular distributions for this study.
Density of Structural Planes (Section 2.2.3)
The discussion on the density of structural planes is informative. However, adding information on how these density measurements can influence the overall modeling and simulation of rock masses would provide a more comprehensive understanding​​.
Modeling Based on DFN-DEM Approach (Sections 3 and 3.1)
This section effectively describes the modeling process. To strengthen this section, consider discussing the specific challenges encountered during the modeling process and how they were overcome​​.
It would be insightful to include a subsection on the validation of the model. How do the results compare with empirical data or observations from the field?
Random Fracture Network Simulation Program (Section 3.1)
The development of the "Random Fracture Network Simulation Program" is a key contribution of this paper. Expanding on how this tool can be adapted or used for different types of rock masses or different geological settings would add value to the paper​​.
A brief discussion on the potential for integrating this tool with other simulation software or platforms currently used in geological engineering could provide insights into its practical applications.
In summary, the paper presents a significant contribution to the field of geological engineering. The above suggestions aim to enhance its impact by improving clarity, depth, and applicability.
Reviewer 3 Report
Comments and Suggestions for Authors
The authors present the results of the developing of the numerical approach for the effective generation of equivalent broken rock mass models, allowing for the investigation of mechanical behavior and the determination of REV size of given broken rock masses. The three-dimensional fracture network generation program using the theories of discrete fracture network (DFN) and discrete element method (DEM) are presented here. The numerical solutions were compared with the real state of the broken rock masses. A statistical program is built first to evaluate the geometry parameters of the target rock masses; the parameters are then inputted into a self-development numerical program to generate the discrete fracture network based on the Monte-Carlo method. The performance of the proposed approach is demonstrated through a numerical example, which evaluates the mechanical parameters and REV size of the surrounding rock of the main powerhouse at the Liyang pumped storage power station using cylindrical rock models. The determined REV size is verified by the RMSI that evaluates the REV size using cubic rock model.
Opponent's comments:
The article contains a number of formal errors in notations of numerical relationships. It is necessary to correct these relations and put them in the correct format.
I only mention some mistakes:
- The relation (1) is written incorrectly and next relations for A and B !
- The relation (5) is written incorrectly, this write-up is not acceptable !
- The relation (7) is written incorrectly, this write-up is not acceptable !
- The relation (10) is written incorrectly, this write-up is not acceptable !
- The relation (11 and 12) are written incorrectly, this write-up is not acceptable !
- The relation (13) is written incorrectly, this write-up is not acceptable !
- The relation (16, 17) is written incorrectly, this write-up is not acceptable !
- The relations in the table 1 are written incorrectly, this write-up is not acceptable !
- The relation (18) is written two times in the text!
Reviewer 4 Report
Comments and Suggestions for Authors
It is an interesting but confusing article. I can't find the practical contribution. Nor is the contribution to the state of knowledge on the subject clear. Numerical simulations oversimplify reality and micromechanical aspects were not analyzed (e.g., porosity, mineralogical composition, etc.). The experimental phase and resistance tests should be described in more detail.
Round 2
Reviewer 4 Report
Comments and Suggestions for Authors
The authors responded to my comments. It would have been interesting to carry out full-scale tests, however, they carried out multiple simulations on the topic of study. I think the manuscript can be published.